# Assessing the Spatial Agricultural Land Use Transition in Thiès Region, Senegal, and Its Potential Driving Factors

Bonoua Faye [1] , Guoming Du [1,2,]*, Edmée Mbaye [3] , Chang'an Liang [1], Tidiane Sané [4] and Ruhao Xue [1]

1   School of Economics and Management, Northeast Agricultural University, Harbin 150030, China
2   School of Public Administration and Law, Northeast Agricultural University, Harbin 150030, China
3   Department of Geography, Cheikh Anta Diop University, Dakar 5003, Senegal
4   Department of Geography, UFR Sciences and Technologies, Assane SECK University, Ziguinchor 523, Senegal
*   Correspondence: duguoming@neau.edu.cn

**Abstract:** The agricultural land use transition (ALUT) assessment can be a prominent tool for comprehensively implementing suitable agricultural land use and agricultural development in Senegal. Based on remote sensing and survey data, this investigation aimed to simultaneously assess the geographical dispersion of ALUT and its probable mechanisms and determine the agricultural land functions in the Thiès region. Through ArcGIS and ENVI software, the remote sensing data of 2000, 2010, and 2020, the transfer matrix method and a spatial index calculation were used to characterize the ALUT. Then, the mixed linear regression model was constructed to determine the relationship between ALUT and its potential driving factors. The main results show that ALUT was about $-588.66$ km$^2$. Regarding spatial distribution, a positive net ALUT was experienced in the north-west department of Tivaouane; conversely, a negative transition was noted in the southern Mbour department. The agricultural land per capita (0.37 ha/per) and the per capita agricultural income (USD \$167.18) were unsatisfactory, and only 59.49% of the respondents frequently used fertilizers for production. The linear regression model showed that rainfall variability, research and development, soil salinization, and land tenure were significant at 1% ($p < 0.01$) with agricultural land change, living security, and ecological functions. Parallelly, transportation facilities ($p < 0.01$) and agricultural investment ($p < 0.01$) were also significant with agricultural land morphology change, while population growth ($p < 0.1$) was only correlated with agricultural land morphology change. These factors reflect the farmer's income and often induce land abandonment and degradation of agricultural land. Consequentially, the ALUT in the Thiès region revealed several insights, such as the need to strengthen land use reforms and research and development. Therefore, agricultural land use is impacted by many fields that require an inter-discipline research method for practical and balanced use. Such endeavors could start with reconciling agricultural development and land conservation.

**Keywords:** Thiès; Senegal; agricultural land use transition; social survey; agricultural land functions; potential driving factors

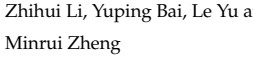



## 1. Introduction

Soil is a non-renewable resource, and it should be maintained to support food production and, thus, ensure food security [1]. So, implementing suitable agricultural land management and analyzing the farmers' knowledge is essential for agricultural land. In recent years, various questions regarding ALUT, how to coordinate population growth, urban expansion, and agricultural production to ensure food supply has raised huge concerns worldwide [2]. This situation exists because the agricultural land use resource is an indispensable production factor for national economic growth and farmer households. Alternatively, it is emphasized that various factors constantly negatively affect the effective use of agricultural land. The ALUT process is often non-linear [3] and different from one country to another. It may revolve around two interrelated questions: the physical

environment and socio-economic driving factors [2,4,5], and it is associated with other societal and biophysical system changes [6,7] Land-use transforms, such as converting semi-natural grasslands to agriculture or high-intensity pastures, affect biodiversity, ecosystem functions, and services [4]. In India, increasing food demands are causing rapid changes in farming systems, often involving intensified land use [5]. Simultaneously, in China, economic change generally leads to increased demand for land and changes in land utilization patterns [7]. In Japan, in the population growth phase, geographical conditions such as the ratio of central farmlands, the fertility of farmlands, and the ratio of past wetland farmlands contributed significantly to explaining farmland abandonment [8]. In addition, some research identified an increase in urban population (4%), low agricultural returns (29%), demand for housing (16%), and weak, ineffective land institutions (13%) as the major causes of urban sprawl in Wote town, Kenya [9]. Therefore, the ALUT process and its effect depend on the socio-economic and geographical context. So, whatever the context and region, population growth can negatively impact agricultural land morphology and reduce potential agricultural functions.

In the Thiès region, the ALUT process seems complex. The Thiès region accounted for about 1,788,864 inhabitants in 2013, with a projected 2,464,554 inhabitants by 2025, according to the National Agency for Statistics and Demography of Senegal (ANSD). Regarding infrastructure, during the last decade, the Thiès region has been chosen to host significant structuring projects such as the Blaise Diagne International Airport and the Special Integrated Economic Zone. This situation led to an increase in land demand. For instance, between 2000 and 2020, the construction land temporal evolution reached about 8.56%. From then on, extensive scientific studies have assessed and highlighted the impact of urbanization and population growth on ALUT. For example, in a European context, land take driven by urbanization has severely impacted the agricultural sector's capabilities [10]. The Groundnut Basin of Senegal (including our study area) is confronted with chemical and physical-biological degradation, which has become more intense [11]. Consequently, environmental issues also severely affect the process of ALUT, and in this context, agricultural land may be converted to residential use and later abandoned [12].

In addition to these issues, in Senegal, state legislation and traditional customary rights coexist in a duality that defines agricultural land administration. A weak land policy implementation leads to inappropriate agricultural land use [2]. Between 2017 and 2019, about 88% of farmers' land did not have a formal document from the state. Accordingly, in this context, urbanization is another major social, economic, and demographic trend with consequences for the structure and functions of agricultural landscapes [13]. An uncontrolled ALUT may threaten the food supply's capacity and limit the continuous improvement of the farmers' living security. As a result, human activities and natural factors have the potential to stimulate new land use while also reshaping the morphology and functions of agricultural land and directly affecting household income. Consequently, understanding the dynamics, evolution, and existing driving factors of agricultural land morphology and functions is crucial to formulating national food security objectives. In sum, as the spatial carrier of the agricultural system, proper land management is vital in promoting socio-economic development, living security, and production functions in particular.

In an agricultural system, land capacity production may be defined as a process through which a farmer transforms inputs into outputs. Grain production, living security, and eco-environmental functions are the core functions of agricultural land use [14]. The evaluation of production functions considers two elements: agricultural production capacity and industrial development vitality [15]. In developing countries such as Senegal, the agricultural production capacity and industrial development are still insufficient. In 2016, Nijbroek et al. highlighted that crop production per capita remains at 1960 levels in Africa, while the agriculture sector accounts for 65% of full-time employment [16]. In that setting, like in many other African countries, agricultural land functions, namely grain production, are challenging in Senegal. Parallelly, agricultural input, namely fertilizers, is still very weak compared to industrial countries. In Senegal, farmers use 25.9 kg of fertilizer per

hectare [17], whereas they use 383.32 kg per hectare in China, according to the World Bank in 2020. Hence, it is essential to note that the overuse of chemical fertilizers has caused various challenges, such as soil degradation [18]. In the Groundnut Basin of Senegal (including the Thiès region), the soils are impoverished, restructured, and chemically exhausted by wind, water erosion, and ongoing droughts [11]. The relation between production and ecological functions has a trade-off effect, and the significance gradually increases [15]. In contrast, the depreciation of ecological functions directly impacts agricultural production. So, addressing societal problems like enhancing agricultural production is a big challenge for agricultural land functions in the Thiès region.

Agricultural land in Senegal is passed down from generation to generation. So, the agricultural land fragmentation due to inheritance modified the agricultural land morphology. From then on, technological, socio-economic, and policy targets were urgent for sustainable agricultural land management. Another fact is that rainfall variability affects agricultural production [19], inducing the depreciation of agricultural land functions through the decreased yield, which leads to land abandonment and migration. Population mobility is the choice of the rural population to move between urban and rural areas to obtain employment opportunities and increase income [20]. Accordingly, the imbalance between urban and rural developments will become a significant obstacle [21]. In addition, the COVID-19 pandemic has posed enormous obstacles to the agri-food system, including a lack of inputs and technical support [22].

What does existing research reveal about factors influencing ALUT and its functions, and the circumstances under which land policies and services act to mitigate the loss of agricultural land use? What are the best methodologies for making a holistic analysis for evaluating the relationship between agricultural land use and its influencing factors? What is the range of challenges survey data may reveal for more acknowledging the mean driving factors of ALUT? What challenges may survey data show to understand better the main driving factors of ALUT? However, given this context and questions, it is worth noting that a thorough examination of agricultural land morphology is required for Senegal. In other words, Senegal lacks relevant studies at the regional level concerning the link between the ALUT and agricultural land functions and, therefore, requires a holistic investigation. Previous studies often analyzed the issues of agricultural land change and agricultural development separately. The role of fertilizer subsidies in agricultural productivity [17] and the relationship between rainfall, sown land area, and yield [23] are among Senegal's most significant investigations. In addition to these studies, the heterogeneity in credit constraints and the relationship between landscape and urban planning [24] were among the most focused on land use and agricultural development issues. Then, an integrated analysis of ALUT and its functions, such as remote sensing and survey data, is scarce. So, Senegal has inadequately comprehended and researched multifunctional land use and agriculture. Said another way, there is still a lack of qualitative and quantitative analyses of ALUT and its functions. As a result, acknowledging the fundamental driving factors of ALUT while also assessing agricultural land functions in the Thiès region from the perspective of the social survey may provide significant insights into Senegal's agricultural development.

However, agricultural land use evolution depends on several factors, such as agricultural investment and rainfall variability. In addition, political driving factors like weak or unclear land tenure can hide agricultural development. As a result, there is a close nexus between spatial ALUT, functions, and driver factors. Following this ascertainment and regarding the research gaps highlighted above, addressing holistic research methodology to analyze the ALUT process is required in the Thiès region in Senegal. From then on, this investigation chooses the Thiès region as a case study to assess ALUT and its relevant influence factors from the regional perspective. Our specific objectives are: (1) to explore the spatial characteristics of ALUT; (2) based on social survey data, to determine the agricultural land use functions; and (3) through multiple regression analysis models, the study assesses the potential influencing factors of ALUT.

## 2. Presentation of the Thiès Region

The spatial extent of the Thiès region is between 10°44′46″ and 10°52′46″ north latitude and 78°39′11″ and 78°44′13″ west longitude. The Thiès region is an agricultural zone where agriculture, especially groundnuts and vegetables, became essential to Senegal's economy. Regarding land area, it is one of the smallest regions in Senegal, at about 6669.6 km² or 3.35% of the total land area of Senegal. The Thiès region had 2,162,831 inhabitants in 2020, according to ANSD. From the perspective of agricultural functions, namely crop production, the main crop types are peanut, maize, millet, sorghum, and cowpea. The agricultural data from ANSD show that the sown land area of these main crop types above represented about 266,668.24 hectares in 2020. In the same period, the agricultural production of these crops was around 25,3784.08 tons, according to ANSD statistics collected in October 2022.

From the spatial land use morphology perspective, Figure 1 shows that agricultural land represented 56.94% in 2000, compared to 48.4% in 2020. Grassland represented about 13.61% in 2000 and 18.4% in 2020. In the same sense, construction land represented 3.5% in 2020, conversely, 1.24% in 2000. So, agricultural and grassland represented the most significant agricultural land dominant morphology, accounting for 59.46% of the total land area in 2020. In sum, there are two significate findings worth mentioning. Agricultural land (−0.71%), ecological land (−0.25%), and wetland (−1.22.3%) decreased significantly between 2000 and 2020. Conversely, grassland (1.66%), unused land (0.79%), and construction land (8.56%) increased during the same period.

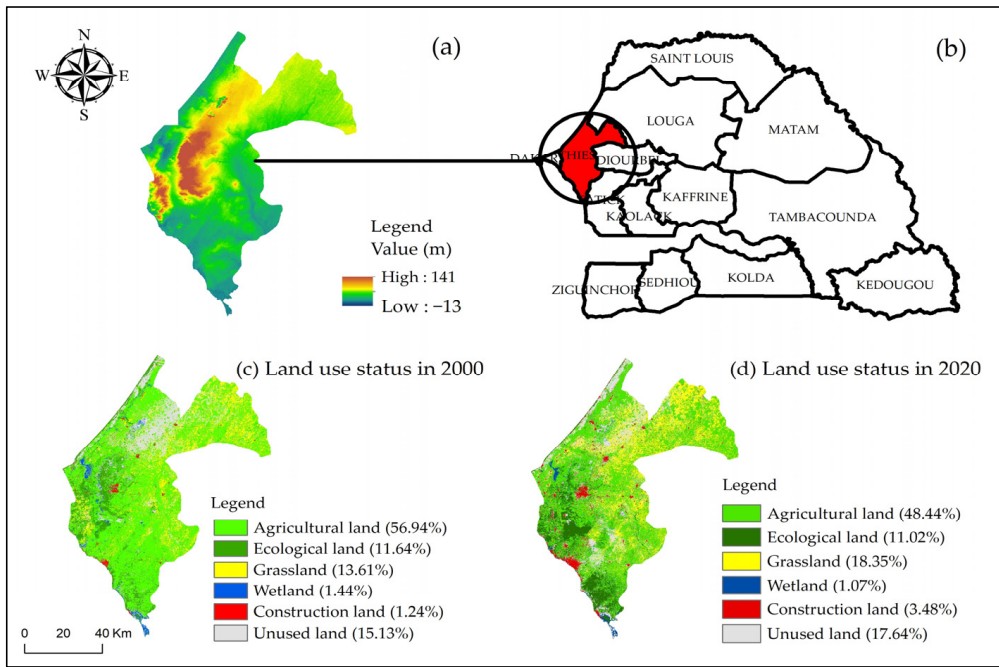

**Figure 1.** Description of the main features of the study area. (**a**) Digital Elevation Model (DEM); (**b**) Senegal's administrative limits regions, (**c**,**d**) the land use status in 2000 and 2020, respectively.

The topography is flat except for the "Plateau of Thiès," which culminates at 141 m altitude. The temperatures are generally high, and the annual temperature cycle is complex. The maximum temperature is 33.2 degrees. In addition, the interannual evolution of rainfall shows that the average rainfall was about 461.65 mm from 2000 to 2020, according to the data collected by the National Agency of Civil Aviation and Meteorology (ANACIM). The main soil properties are ferruginous tropical sandy soils and slight leaching [24].

## 3. Materials and Research Methodology

### 3.1. Data Sources

3.1.1. Remote Sensing Data

The shapefile data corresponding to the limit of the administrative communes was collected from the Ecological Monitoring Centre (CSE) in Senegal. However, to achieve the research's aim, this paper takes all 31 administrative communes as a scale to analyze the spatial-temporal evolution of ALUT and its characteristics from 2000 to 2020. The remote sensing data came from various satellites, including Landsat 7 ETM, Landsat 5, and Landsat 8 OLI (Table 1). All the remote sensing images were obtained from the United States Geological Survey (USGS) website with a spatial resolution of 30 m (http://earthexplorer.usgs.gov/ (access date: kindly see Table 1)).

**Table 1.** List the satellite images used for the study and their information.

|      | Acquisition Date | Image Types | WRS Path/Row | Proportion of Cloud % | Collected Date |
|------|------------------|-------------|--------------|------------------------|----------------|
| 2000 | 11 November | Landsat 7 ETM + C1-L1 | 205/50 | 1 | 31 August 2022 |
|      | 11 November | Landsat 7 ETM + C1 | 205/49 | 7 | 31 August 2022 |
| 2010 | 25 October | Landsat 5 TM C2-L1 | 205/50 | 0 | 6 July 2022 |
|      | 25 October | Landsat 5 TM C2-L1 | 205/49 | 6 | 6 July 2022 |
| 2020 | 20 October | Landsat 8 OLI-C2-L1 | 205/50 | 1.94 | 22 August 2022 |
|      | 20 October | Landsat 8 OLI-C2-L1 | 205/49 | 1.35 | 22 August 2022 |

The collection period of remote sensing images is essential to determine agricultural land accurately. Indeed, Senegal has two main seasons that determine the climatic regime: a dry season from November to April–May and a rainy season from May–June to October, depending on the geographical location [25]. The Thiès region is one part of the ground basin in Senegal, where the wintering season extends from June to October [26], coinciding with our study area's rainy season. However, to maximize the features of agricultural land use, we chose the winter months to reduce the negative impact of clouds and seasonal variation. Therefore, according to Feteri et al., the selection of Landsat images was mainly based on availability, cloud cover percentage, and correspondence [27]. Due to these constraints, the Landsat images were collected between October and November.

3.1.2. Social Survey Data

A comprehensive questionnaire for a social survey was designed to collect information about the potential factors affecting the ALUT from the farmers in 11 administrative communes in the Thiès region. Specifically, from a spatial point of view, 210 questionnaires were collected in four communes in the Tivaouane department (North). In Thiès' department (in the center), four communes were also investigated for 190 questionnaires. Lastly, three communes were selected in the Mbour department (South) for 200 questionnaires. In total, 600 questionnaires were primarily collected in October 2022. In addition to this social survey, a face-to-face field interview was conducted with the commune administrators. Globally, the survey questionnaire was composed of four sections. Only the third section relates to farmers' perceptions of the ALUT's potential driving factors, which this paper explores. The CommCare HQ software was used as a tool for collecting data. The face-to-face method was adopted by paying attention to ethical considerations such as sensitive responses like agricultural income. The data screening process shows that 15 questionnaires were discarded due to a lack of logic. For this reason, 585 completed questionnaires were used in the following analysis. The sampling strategy for respondents was built on a stratified random sampling design.

### 3.2. Research Methodology

Given the study area's size, two Landsat images were collected yearly. However, due to the characteristics of the remote sensing data, pre-processing is necessary to have more clarity. Therefore, several steps have been taken. First, to optimize the quality of the images, the layers were re-projected according to the reference projection system of the study area, which is World Geodetic System (WGS)_1984_Complex_UTM_Zone_28N (EPSG:31028). This projection allows us to obtain expected results between the processed images. Then, we resampled the remote sensing images to 50 m, the standard resolution for all images [28]. Second, geometric correction, such as atmospheric correction, gap fill in Landsat 7 ETM, and image mosaicking through ENVI software were performed. After this step, the supervised classification is chosen for this study, and training samples are selected for each land cover class. Human–computer interaction interpretation methods extracted land use information from the remote sensing image data.

From then on, it is important to note that land use classification systems vary with the purpose and context of their use [29]. Consequently, using the classification system of Anderson JR and al. as a reference [30], we have reclassified the land use types into six categories: (1) agricultural land, (2) ecological land, (3) grassland, (4) wetland, (5) construction land, and (6) unused land. Additionally, it is essential to highlight that, during the classification of land use types, Google Earth played a significant role in identifying the unclear characteristics of certain land use morphologies. In addition, the remote sensing images were clipped according to the size of the research area. Finally, after the raster conversion to polygons, we used the ArcGIS 10.6 platform to determine land use types' statistics and quantify the ALUT for different periods.

An accurate assessment is essential for processing land use change analysis and classification [31]. However, the overall accuracy values based on the post-classified images generated in 2000, 2010, and 2020 differed yearly. For instance, the least accurate year is 2020, with 0.91. However, the overall accuracy for our study period was 0.93. Additionally, the kappa coefficient was about 89.05%, indicating that the simulation results have high consistency and accuracy with the actual LULC distribution [32] because an overall standard accuracy for LULC classification is estimated to be approximately 85% [30].

#### 3.2.1. Calculate the Method of Characterizing ALUT

*a.    Tracing the sources and flows of agricultural land use*

Tracing the sources and flows of agricultural land use can assist in determining how agricultural land is lost or gained from other types of land (transfer in or out) [33]. So, this process has followed many steps (Table 2). The computed transition matrix consists of rows that display categories at period *T1* and columns that display types at the period *T2*. The notation *Aij* is the land area that experiences a transition from category *i* to category *j*. The diagonal elements (i.e., *Aii*) indicate the area of the land that shows the persistence of category *i*. Entries off the diagonal indicate a transition from category *i* to a different category *j*. The area of the land in category *i* in period *T1* (*Ai+*) is the sum of *Aij* overall *j*. Similarly, the land area in category *j* in the period *T2* (*A + j*) is the sum of *Aij* overall *i*. The losses (*Ai+–Aii*) were calculated as the differences between row totals and persistence. The gains (*A + i–Aii*) were calculated as the differences between the column totals and persistence [28,29].

The spatial evolution of land use change is frequently characterized by amplitude. In this study, the amplitude index of agricultural land net transition evolution was mainly characterized by the value of change in the quantitative transition of agricultural land. It was measured according to the land area of each commune. The equation below determines the spatial index of agricultural land use change [34]:

$$B_{it+n} = \left[ \frac{(U_{it+n} - U_{it)}}{T} * 100 \right] \tag{1}$$

where $B_{it+n}$ is the annual expansion intensity of spatial unit $i$; $U_{it+n}$ is land use type area at the spatial unit $i$ at time $t + n$; $U_{it}$ is land use area at the spatial unit $i$ at time $t$; and $T$ is the land area of at the spatial unit $i$.

**Table 2.** A sample transfer matrix of land use change method.

| | | *Time 2 (T2)* | | | | *A + i* | *Loss in Land Area* |
|---|---|---|---|---|---|---|---|
| | | *L1* | *L2* | ... | *Ln* | | |
| | *L1* | *A11* | *A12* | ... | *A1n* | *A1+* | *A1+–A11* |
| | *L2* | *A21* | *A22* | ... | *A2n* | *A2+* | *A2+–A22* |
| *Time 1 (T1)* | ... | ... | ... | ... | ... | ... | ... |
| | *Ln* | *An1* | *An2* | ... | *Ann* | *An+* | *An+–Ann* |
| | *A + i* | *A + 1* | *A + 2* | ... | *A + n* | | |
| | *Gain in land area* | *A + 1–A11* | *A + 2–A22* | ... | *A + n–Ann* | | |

b.    *Kernel density estimation (KDE)*

Kernel density estimation reflects the spatial distribution density and changing trend of point groups [35]. This study uses kernel density to show the spatial agricultural land use transition in the Thiès region from 2000 to 2020. KDE is one of the statistical methods of nonparametric density estimation, modeled as follows in the equation above:

$$f(x, y) = \frac{1}{nh^2} \sum_{i=1}^{n} k\left(\frac{d_i}{n}\right) \tag{2}$$

where $f(x, y)$ is the density estimation located at $(x, y)$ position, $n$ is the observation numbers, $h$ is the bandwidth or smoothing parameter, $k$ is the kernel function, and $d_i$ is the distance from position $(x, y)$ to observation position $i$.

c.    *Local spatial autocorrelation*

An exploratory spatial data analysis can perform a correlation and aggregation analysis of spatial cluster data, effectively verifying the spatial clustering characteristics of regional agricultural land use transition. Two types of autocorrelation coefficients are usually used for this measurement. The first is the global spatial autocorrelation coefficient: the distribution of the Moran scatter plot is used to show the spatial correlation of ALUT in the study area. The expression is:

$$I = \frac{n \sum_{i=j}^{n} \sum_{j=1}^{n} (x_i - x)(x_j - x)}{\sum_{i=1}^{n} \sum_{j=1}^{n} w_{ij} \sum_{i=j}^{n} (x_i - x)^2} \tag{3}$$

where $I$ is the global Moran index, $x_i$ and $x_j$ are the agricultural land transition index in cities $i$ and $j$, respectively, $x$ represents the average of the agricultural land transition indices, and $W_{ij}$ is the spatial weight matrix. This study used a spatial adjacency matrix constructed by ArcGis software. The value of $I$ is $[-1, 1]$. When $I = 0$, this indicates that the space is not autocorrelated; when $I > 0$, this indicates a positive correlation; and when $I < 0$, this indicates a negative correlation. The closer the absolute value of $I$ is to $1$, the greater the degree of clustering and the spatial correlation. The second type is the local spatial autocorrelation coefficient: it can use a LISA graph to check the heterogeneity of the data calculation and reveal the correlation degree of the attribute values between spatial units and adjacent units. The formula is as follows:

$$I_i = \frac{n(x_1 - x) \sum_{j=1}^{n} w_{ij}(x_j - x)}{\sum_{i=1}^{n} (x_i - x)^2} \tag{4}$$

when $I_i > 0$, high-high/low-low means that the spatial unit value is higher/lower than all the surrounding units and that the integrated spatial difference is smaller. When $I_i < 0$, then

low-high/high-low means that the lower/higher spatial unit value is higher/lower than the surrounding units and that the integrated spatial difference is smaller.

### 3.2.2. Agricultural Land Morphology and Functions and Its Driving Factors
Determining Relevant Fundamental Influencing Factors

Identifying major underlying factors of ALUT is important for developing countries to meet a comprehensive land structure and management. The African continent is growing in importance with climate change and population pressure on land [36]. As a result, the African continent development process will face many challenges for proper land management. So, complex driving factors, such as socio-economic [37] and natural environment [38], were selected as a reference to evaluate the main agricultural land use transition factors. As can be seen in Table 3, the social and economic variables chosen include (1) population growth and (2) transportation facilities. We assume that (3) a farm labor shortage and (4) a lack of agricultural investment facilitates farmland abandonment. Accordingly, these variables were added to the socio-economic variables to make the research more understandable.

**Table 3.** Description of the potential driving force of agricultural land use transition.

| N. | Variables | Main Score Values (%) | Coding |
|----|-----------|----------------------|--------|
| 1 | Population growth | With score 2 = 52.48% | |
| 2 | Transportation facilities | With score 2 = 48.38% | |
| 3 | Farmers' labor force | With score 2 = 56.41% | |
| 4 | Lack of agricultural investment | With score 2 = 62.91% | 1 = strongly agree; 2 = agree; 3 = neutral; 4 = disagree; and 5 = strongly disagree |
| 5 | Rainfall variability | With score 2 = 54.87% | |
| 6 | Wind erosion | With score 2 = 54.53% | |
| 7 | Soil salinization | With score 2 = 58.46% | |
| 8 | Hydric erosion | With score 2 = 58.63% | |
| 9 | Research and development | With score 2 = 32.82% | |
| 10 | Number of plots had a formal document | With score 2 = 81.37% | 1 = yes, 2 = no |

Prior work has highlighted that precipitation (rain, snow, etc.) and temperature determine the potential distribution of terrestrial vegetation and constitute the principal factors in the genesis and evolution of soil [39]. The study area has a climate difference, such as rainfall [23]. The average rainfall is the main factor for agricultural production and determines the evolution of the sown land area [19]. So, our investigation considers (5) rainfall variability and (6) wind erosion as factors that may affect ALUT. In addition, (7) soil salinization and (8) hydric erosion were significant variables given the topography in the study area. Regarding political driving factors, (9) research and development and (10) land tenure was selected. Globally, given the socio-economic and natural change complexity in the Thiès region, ten fundamental explanatory variables were chosen for this investigation.

### Determinate the Indexes of Agricultural land Functions

This study selects and defines agricultural land functions and morphology for deep analysis. Therefore, we choose three-level functions. The first is the sown land area. In Senegal, the sown land area evolution depends on several factors, such as rainfall variability, agricultural investment, labor force, etc. The lack of these factors may lead to a decrease in sown land area, or the depreciation of agricultural production and sown land led to agricultural land abandonment. Agricultural land abandonment is increasingly a global land-cover change phenomenon with substantial implications for the environment and societal well-being, including agricultural landscapes [40]. Therefore, agricultural land's spatial

structure will change in this context. So, in this study, we chose the farmers' sown land and agricultural land abandonment status scenario to assess the nexus between agricultural land morphology evolution and its driving factors. Secondly, food insecurity remains a significant issue for rural communities in Senegal [41]. Full exploitation of agricultural land may effectively generate rather good income for farmers, at the same time ensuring their living conditions. Since then, living security and ALUT in developing countries like Senegal have been inextricably linked. So, this study defined the living security function based on agricultural land and agricultural income per capita as the explained variables. Third, eco-environmental aspects maintain the agricultural land's function and are also essential for fighting against food insecurity. In other words, agricultural land protection and quality improvement have become unavoidable prerequisites for reducing ecological and environmental pressures and ensuring long-term agricultural development [42]. From this knowledge, the level of agricultural land protection and the frequent use of fertilizer are selected to understand better the characteristics of ALUT in the Thiès region.

The multiple regression model is based on the assumption that there is a linear relationship between the dependent and independent variables. This research used the multiple regression model to estimate the relationship between the variables to understand the farmer's perception of the potential agricultural driving factors. Using the equation below, the Tanagra software 14.41 served as a tool to compute the results:

$$y_i = \beta_\theta + \beta_1 x_{i1} + \beta_2 x_{i2+} \beta_3 x_{i3} + \cdots + \beta_p x_{ip} + \in \tag{5}$$

where for $i = n$ observation: $y_i$ = dependent variable. This study referred to the variables in Table 4: $x_i$ = Explanation variables. This study referred to the variables in Table 3: $\beta_\theta$ = y-intercept (constant term); $\beta_p$ = slop coefficients for each explanatory variable; $\in$ = the model's error term (also known as the residuals).

**Table 4.** The evaluation indexes of agricultural land use functions (Survey data, October 2022).

| N | First Level | Second Level | Unit | Description and Calculation Method | Average/ Percentage |
|---|---|---|---|---|---|
| 1 | Agricultural land morphology | Sown land status | Frequency | Respondent's land area where diminish | 76.07% |
| | | Land abandonment status | Frequency | Respondents have abandoned plots of land | 42.05% |
| 2 | Living security function | Agricultural land per capita | ha/pers | Total sown land area reported to the family size | 0.37 |
| | | Agricultural income per capita | USA$ | Total income reported to the family size (1 USD = 617,50 XOF 12/12/2022) | 167.18 |
| 3 | Eco-environmental aspects maintain the agricultural land's function | Situation or the level of agricultural land conservation | Frequency | The number of farmers who have used any method of fighting against agricultural land degradation | 55.38% |
| | | The intense use of fertilizer | Frequency | Number of farmers using fertilizer regularly for production | 59.49% |

## 4. Results and Discussion

### 4.1. Spatial Distribution Characteristics of ALUT

#### 4.1.1. Quantify the Net Transition of Agricultural Land Use

Agricultural land quality is directly related to household grain production. Agricultural land use decreased by −566.80 km² from 2000 to 2020 (Table 5 and Figure 2). In this amount, grassland gained about 315.90 km² or 4.736%. In the same line, urbanization gained about 148.95 km², for a spatial index of 2.23%. Accordingly, it is essential to note that construction land has increased rapidly compared to other regions of Senegal. For

instance, the urban built-up evolution represented about 0.93% in the Diourbel region compared to 7.95% in the Thiès region from 2009 to 2018 [43]. At the international level, in northern China, 81.6% of the land occupied by the expansion of rural settlements came from cultivated land between 2000 and 2020 [44]. From then on, we argue that worldwide urbanization has dramatically changed agricultural land structures [45]. Accordingly, the relationship between socio-economic development and the need for new land became one of the main concerns in the ALUT. Following agricultural land use, there is ecological land ($-41.01$ km$^2$) and wetland ($-24.56$ km$^2$), which decreased in the area from 2000 to 2020. So, this scenario may be emphasized in the future because previous studies have highlighted that the urban-cropland changes from 2015 to 2050 will have significant indirect impacts on forest and grassland landscapes [46]. According to ANDS, the exploitation of forest resources plays a central role in Senegal's economy, and the degradation of ecological areas is becoming increasingly alarming. In that sitting, the study found that agricultural land use declined, and grassland expansion was the primary land use type that caused the ALUT.

**Table 5.** Results of the transfer matrix of land use transition (km$^2$) in the Thiès region, 2000 to 2020.

| Land Use Types in 2000 | Land Use Types in 2020 | | | | | | Total—2000 | Loss |
|---|---|---|---|---|---|---|---|---|
| | Agricultural Land | Construction Land | Ecological Land | Grass Land | Unused Land | Wet Land | | |
| Agricultural land | 2178.49 | 87.18 | 380.81 | 601.80 | 540.17 | 8.82 | 3797.27 | 1618.7 |
| Construction land | 10.44 | 59.68 | 1.14 | 1.91 | 9.46 | 0.24 | 82.87 | 23.19 |
| Ecological land | 335.48 | 47.29 | 282.26 | 5.26 | 98.77 | 6.94 | 776.00 | 493.75 |
| Grassland | 411.70 | 14.95 | 16.73 | 360.66 | 102.51 | 1.25 | 907.81 | 547.15 |
| Unused land | 266.79 | 20.58 | 40.35 | 253.86 | 403.56 | 23.57 | 1008.71 | 605.14 |
| Wetland | 27.57 | 2.14 | 13.71 | 0.23 | 21.74 | 30.46 | 95.85 | 65.38 |
| Total—2020 | 3230.48 | 231.82 | 734.99 | 1223.71 | 1176.21 | 71.29 | 6669.51 | |
| Gain | 1051.98 | 172.14 | 452.73 | 863.05 | 772.65 | 40.82 | x | x |
| Total shift | −566.80 | 148.95 | −41.01 | 315.90 | 167.51 | −55.02 | x | x |
| Spatial index (%) | −8.498 | 2.233 | −0.615 | 4.736 | 2.512 | −0.368 | x | x |

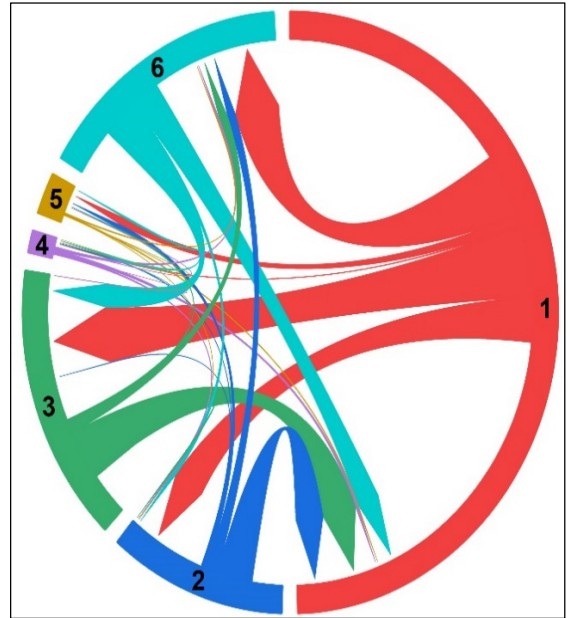
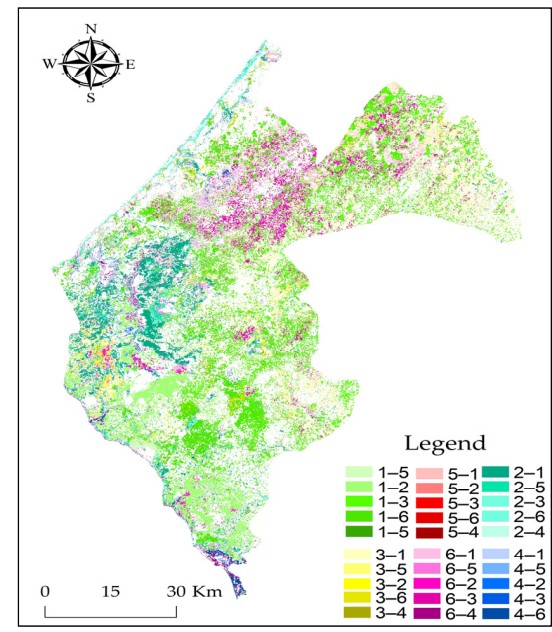

**Figure 2.** Characteristics of transition of agricultural land use from 2000 to 2020: (1) agricultural land; (2) ecological land; (3) grassland; (4) wetland; (5) construction land; and (6) unused land.

4.1.2. The Spatial Variation of the Nuclear Density of ALUT

Kernel Density Estimation (KDE) is a critical approach to analyzing the spatial distribution of point features [47]. The results of this study show a relatively high degree of spatial heterogeneity in ALUT (Figure 3). Land is the spatial carrier of anthropogenic activities [38], but agricultural land use is one of the land use types whose conservation is affected by human activities. From a spatial point of view, between 2000 and 2010, the negative spatial ALUT was localized in the northern Taiba Ndiaye and Meouane communes. In this zone, the negative spatial agricultural land net conversion reached −0.70. This situation may be explained by rainfall variability. According to the National Agency for Civil Aviation and Meteorology (ANACIM), the mean rainfall was about 607.9 mm in 2000 and 577.7 mm in 2020, with approximately 127.4 standard deviations. Consequently, the rainfall variability seems to impact the evolution of the sown land area. In the same sense, agricultural land use increased in our study area's western and eastern parts during the same period, and the maximum value reached 0.44. The Mont-Rolland commune in the West and the Fissel commune in the East can serve as examples.

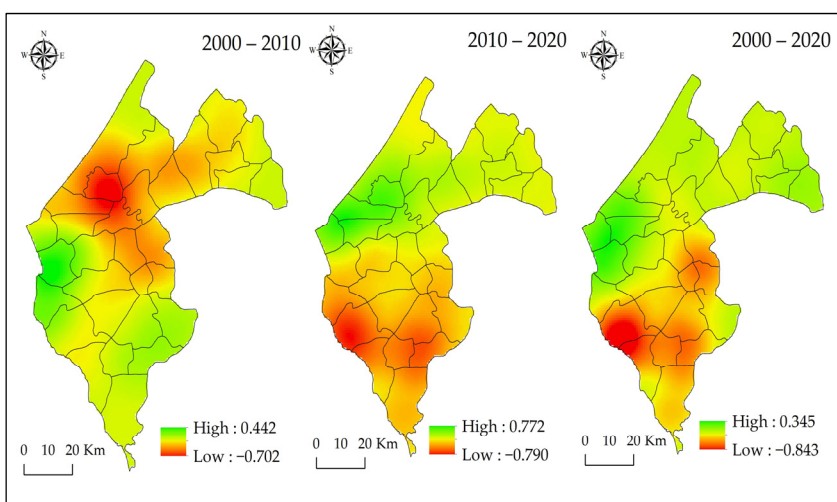

**Figure 3.** Spatial nuclear density of agricultural land use net transition in the Thiès region, 2000–2020.

The period 2010–2020 and the study periods (2000–2020) are similar. Therefore, in contrast to the first period, negative ALUT was clustered in the southern part of the study area, specifically in the communes of Sindia and Malicounda. From the point of view of scale level, the agricultural land use net transition decreased in the Mbour department in the south. Conversely, it increased in the Tivaouane department in the north. So, according to ANSD in 2019, the Thiès' regional urbanization rate represented 51.7%. Then, the Thiès region has served as a secondary region of Senegal to promote socio-economic development [43] with the connection of the highway network such as the Thiès-Touba Toll Expressway Project. Many development professionals see urbanization as a problem [48] because previous studies highlighted that increasing urban mobility, such as trams, allowed a further expansion of cities [49]. The department of Mbour shows a highly negative ALUT, and in 2018, it was the most populated, with 37.4% of the total population. So, the scarcity of land in Dakar, the capital of Senegal [50], reflects directly on the spatial land structure of the Thiès region. Consequently, like in China, economic progress generally leads to a rising need for space and changes in land utilization patterns [7] in the Thiès region. Regarding infrastructure, this area has been chosen to host significant structuring projects such as the Blaise Diagne International Airport [43] and the Special Integrated Economic Zone. In this circumstance, the demand for land became increasingly significant and may explain the pressure on agricultural land use in the south of the Thiès region. Consequently, the impacts of this heavy urbanization on ecosystem services and biodiversity in the Thiès region became unclear.

### 4.1.3. Local Spatial Autocorrelation Analysis

Spatial autocorrelation measures the direction of the linear association between the variables and the degree of intensity of the spatial pattern of a given variable with the same variable but for a defined neighborhood [43]. Hence, from 2000 to 2020, the spatial correlation differed from one period to another. Between 2000 and 2010, the spatial autocorrelation (Moran's I) *p*-value was about 0.617. This situation shows that the geographical distribution of agricultural land use net transition differs. As shown in Figure 4, the spatial correlation was low-high on the coastline, englobed the commune of Darou Khoudous that the agricultural land use net transition was about −20.58%. This coastline is known as "Niaye", the ecological zone, one of Senegal's essential vegetable production areas. The intensity of ALUT may have several causes, and we can note that urbanization and wetlands are steadily converted to agriculture for food security reasons [51]. However, in the Mon-Rolland commune, the agricultural land net transition was high-high. This context has been due to the extension of mixed socio-economic development. During the interview, we note that this commune has received several programs, such as the agri-food project. So, these activities may reflect on agricultural and use extension. In sum, spatial autocorrelation demonstrates multiple processes that generate an association within a locality activity.

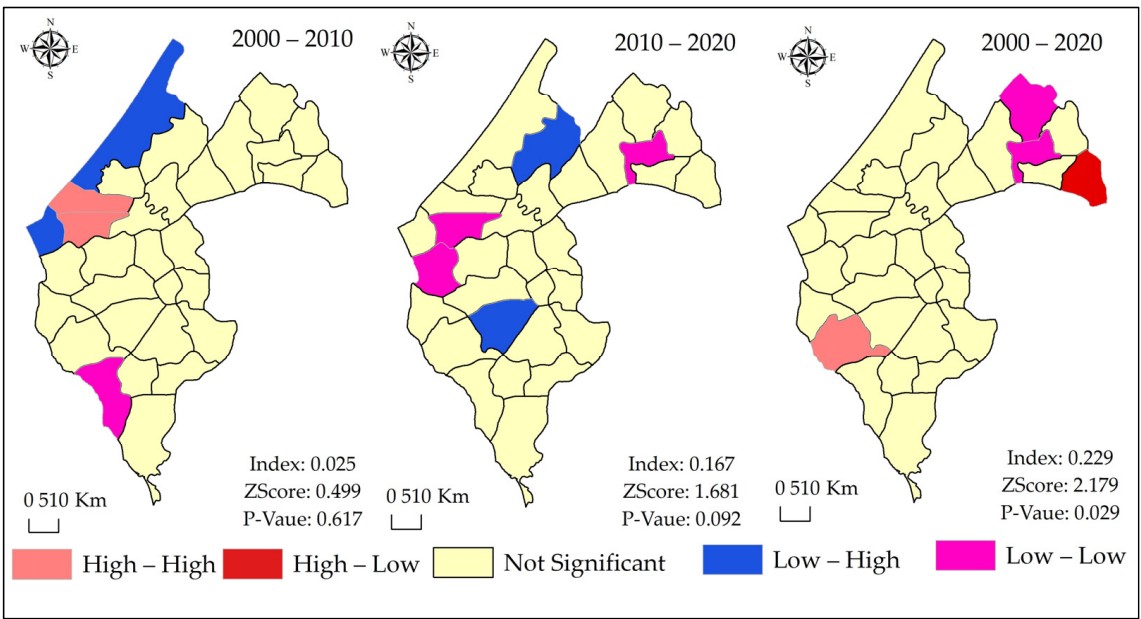

**Figure 4.** Hot pot analysis of agricultural land use transition (ALUT).

From 2010 to 2020, the spatial autocorrelation results were low-high in Meouane in the north and the Tassette commune in the center. Mining activity is essential in the Thiès region. For example, the mining activity in the northern part of the region, in the Taiba Ndiaye commune, strongly impacts the population's socio-economic life [52]. In the center of our study area, namely the commune of Tassette, this situation may be explained by the extension of Thiès town because this region recorded significant urbanization recently. Accordingly, mixed driving factors affect agricultural use in the Thiès region negatively. So, globally, we can note that during the study period (2000 to 2020), the intensity of the agricultural land use net transition was similar (*p*-value 0.029). The most significant was pointed out in the north and south-west.

### 4.2. *Exploring the Potential Influencing Factors of ALUT*

Land use change and agricultural land functions have sparked the attention of researchers and policymakers who focus on sustainable development [53]. This situation was because land provides many functions, such as ecological and economic functions, and the

production of food and fiber [54]. Today the sustainable use of cultivated land is more and more affected by the growing population [2], changing diets [55], and the changing global climate [56]. Therefore, land change and agricultural land's functions are complex.

### 4.2.1. Agricultural Land Morphology Change

In general, the relationship between agricultural land functions and socio-economic, natural, and political driving factors was significant in this study. Previous studies highlighted that urbanization shrinks forests and grassland [57]. From a socio-economic point of view, the sown land area shows a significative correlation with the transportation facilities (0.000386) and population growth (0.050148). In the Thiès region, urbanization represented 1.24% in 2000, compared to 3.48% in 2020. Hence, land use area changes, structural changes, and spatial transfers all impact the number of patches in the landscape and affect the output of ecosystem service functions [58]. Urbanization leads to a continuous loss of agricultural land, both directly in the form of land taken and indirectly through the use of agricultural land for non-productive rural activities like recreation or hobby farming [49].

Consequently, urban sprawl reflects directly on agricultural land use, and peri-urban land is faced with a significant challenge that can affect the spatial transformation of agriculture. Farmers' labor force (0.014133) and agricultural investment (0.066225) also appear in this study as other factors influencing agricultural land morphology—the lack of investment in agriculture is linked to institutional changes [59]. For instance, about 84.4% of the respondents had not received government subsidies. Hence, the lack of labor force and investment led to land abandonment, significantly correlating with agricultural investment (0.002438). This context may imply that the agricultural sector's unrivaled potential to strengthen the socio-economic circumstances of large proportions of the population has not been matched by the level of financial inclusion in the form of credit [60], particularly in terms of facilitating optimal agricultural land exploitation.

Therefore, the socio-economic led to the other challenge because soil salinization significantly correlated with sown land evolution (0.003655) and agricultural land abandonment (0.008956). Climate change can negatively affect crop yields and livestock production, thus threatening food security, especially in a vulnerable continent like Africa [61]. So, it appears evident that natural and climatic factors such as rainfall play an essential role in the agricultural land system on the African continent, Senegal in particular. This study's rainfall variability significantly correlates with sown land evolution (0.005602). According to the findings, increasing agricultural research and development may appear to be a viable solution for optimizing agricultural land use and preventing land abandonment.

### 4.2.2. Living Security Function

The evolution of agricultural land use and living security, such as agricultural income, are closely linked in Senegal. In the Thiès region, the survey data show that the family size of respondents was 9.37 people per household, corresponding to 0.37 hectares, and USD 167.18 in farmer income per person. The ANSD statistics highlight that only 3.7 persons in the family participate in agricultural production and are therefore considered as a farmer labor force. In addition to these results, the main peanut crop types record significantly different yields over the years, also contributing to reducing the potential agricultural income. For example, peanut yield increased by 944 kg per hectare in 2018 compared to 1197 kg per hectare in 2020. So, the agricultural income and yield seem too low to meet adequate living security. This situation led to the decline of agricultural production [62] and, therefore, the decline in agricultural interest.

Urbanization not only increased the demand for agricultural products [14] but induced the loss of agricultural land. In this study, the main factors were rainfall, which can directly link to other subsequence factors such as investment, sown land area, etc. The natural factors, namely rainfall variability, significantly correlate with agricultural land per capita (0.000241) and farmer income per capita (0.000002). These results show that rainfall appears to be the core agricultural production factor in the Thiès region. This situation was

highlighted by previous scientific results that attest the sown land area and agricultural production level depend primarily on rainfall [18]. In the same sense, farmer income per capita (0.041487) has been influenced by transportation facilities. Transportation facilities seem like a double-edged sword. First, the development of roads, such as highways, induces urbanization, which is a potential factor leading to agricultural land loss. Secondly, transportation is essential in rural localization for switching goods from the farm to the markets. Such a company is right for nearby productiveness, enhancing the livelihood opportunities of the local farmers [63]. The survey data highlighted that roads are one of the main barriers for farmers to access the market. Only 8.41% of households have a transport method for moving agricultural products to the market; conversely, the lack of transportation induces the depreciation of the price of agricultural products.

So, transportation facilities and agricultural income are closely linked, and the more their accessibility is convenient, the more agricultural income may increase and vice versa. Exploring the data from ANSD in 2020, we found that about 51.8% of the farmers' plots faced socio-economic or natural factors that hindered their potential production. Accordingly, referring to government funding for the operating and capital expenses associated with conducting research and development programs is still very low [64] for enhancing farmer living security. Ultimately, the Thiès region faces the duality between the need for land to ensure food security but, on the other hand, the land area needs to support socio-economic development in the process. However, urban sprawl and transportation facilities are inseparable. Urbanization also economically squeezes agricultural production, resulting in farmland marginalization and causing serious threats to food security [65].

### 4.2.3. Eco-Environment Maintenance of Agricultural Land Function

Sown land area and living security, such as farmer income, depend on agricultural land quality. A low agricultural investment induces land abandonment that directly impacts agricultural land degradation, which shows a strong relationship with investment (0.009217). In the Thiès region, the rainfall variability determines the farmer's decision on the size of sown land area. This study showed that rainfall variability significantly correlates with agricultural fertilizer ($p < 0.01$). Like transportation facilities, the impact of fertilizer use is double. The overuse of fertilizer leads to land degradation, while proper use of fertilizers can enhance crop yield [66]. Another fact, soil salinization shows a significant correlation and land degradation ($p < 0.01$). Accordingly, agricultural land abandonment facilitates land degradation because it poses an ecological opportunity or threat that depends upon regular management for their existence [67]. Based on this assertion, it is worth noting that the farmers' labor force is a significant factor in the optimum exploitation of sown land areas and in reducing the potential for farmland abandonment. The results show that the agricultural land conversation level was weak. Only 55.38% of respondents used any method of protecting it, and approximately 81.37% of plots did not have formal documents. Then, according to ANSD statistics, about 28.41% of farmer' plot was under conventional sustainable land management. Along the same line, natural resources such as forests continue to be degraded because about 4.23% of the households attest that they have cleared protected forests, and 6.34% have used harmful pesticides on the environment. The above imbalances emphasize risk and uncertainty in grain crop production and suitable agricultural land use in the Thiès region.

In other words, natural environmental and biophysical factors combined with strong population growth and high densities will hurt agricultural land availability. That is why recent calls for transformations in our agricultural landscapes emphasize the importance of agricultural systems that boost ecosystem services for agriculture through environmentally, economically, and socially beneficial practices while also maintaining or increasing productivity [68]. From then on, the stability of agricultural land rights can help to limit land degradation, showing a strong relationship with land tenure ($p < 0.01$). In short, as shown in Table 6, the climate, topography, and socio-economic factors significantly influenced agricultural land morphology and function change. Agricultural land use is currently

the most fundamental land use type, providing essential benefits to humans. Therefore, protected agricultural land faces several challenges in the Thiès region because of weak land policies. It appears the control of agricultural land use against socio-economic, natural, and political factors is insufficient. There is an urgency to ensure land is managed efficiently, equitably, and responsively [69].

**Table 6.** The regression analysis results for agricultural land use and its influencing factors.

| Level I | Level II | Agricultural Land Morphology Change | | Living Security Function | | Eco-Environment Maintenance Agricultural Land Function | |
|---|---|---|---|---|---|---|---|
| | | Sown land Evolution | Agricultural land abandon | Agricultural land per capita | Farmer income per capita | Agricultural conservation level | Agricultural fertilizers used frequently |
| | (Intercept) | 000000 *** | 000000 *** | 000000 *** | 0.000001 *** | 000000 *** | 0.000002 *** |
| Socio-economic driving factors | PG | 0.050148 * | 0.782446 | 0.924369 | 0.949626 | 0.18602 | 0.537107 |
| | TF | 0.000386 *** | 0.072803 * | 0.152944 | 0.041487 ** | 0.054434 * | 0.014309 ** |
| | FLF | 0.014133 ** | 0.675247 | 0.304713 | 0.869623 | 0.015784 ** | 0.073276 * |
| | AI | 0.066225 * | 0.002438 *** | 0.275338 | 0.373963 | 0.009217 *** | 0.969148 |
| Natural and climatic driving factors | RV | 0.005602 *** | 0.052266 * | 0.000241 *** | 0.000002 *** | 0.389152 | 000000 *** |
| | WE | 0.992705 | 0.783634 | 0.599841 | 0.898509 | 0.003984 *** | 0.330631 |
| | SS | 0.003655 *** | 0.008956 *** | 0.492751 | 0.625101 | 000000 *** | 0.018829 ** |
| | HE | 0.555487 | 0.328227 | 0.636611 | 0.584647 | 0.021972 ** | 0.378208 |
| Political driving factors | RD | 0.014066 ** | 000000 *** | 0.212554 | 0.821024 | 000000 *** | 0.000001 *** |
| | LT | 0.000007 *** | 0.029396 ** | 0.907655 | 0.264722 | 000000 *** | 0.04812 ** |
| | Num.Obs. | 585 | 585 | 585 | 585 | 585 | 585 |
| | R2 | 0.144439 | 0.188411 | 0.05354 | 0.063436 | 0.340459 | 0.203325 |
| | R2 Adj. | 0.129534 | 0.174272 | 0.037051 | 0.047119 | 0.328968 | 0.189446 |
| | Sigma error | 0.525725 | 0.448954 | 0.041876 | 139.004495 | 0.804081 | 0.442354 |
| | F-Test (10,574) | 9.6905 (0.000000) | 13.3255 (0.000000) | 3.2471 (0.000432) | 3.8878 (0.000040) | 29.6302 (0.000000) | 14.6495 (0.000000) |

\* $p < 0.1$, \*\* $p < 0.05$, \*\*\* $p < 0.01$. Population Growth (P.G.); Transportation Facilities (T.F); Farmers' labor force (FLF); Agricultural investment (A.I.); Rainfall Variability (R.V.); Wind erosion (WE); Soil salinization (S.S.); Hydric Erosion (HE); Research & Development (R.D.); Land tenure (L.T.).

### 4.3. The Contributions of the Study

As shown at the beginning of this research in Senegal, some shortcomings still exist regarding ALUT and its influencing factors. As a result, this study's methodological contribution is that it used a holistic analysis that included a survey and remote sensing data to understand ALUT trends and their influencing factors. Then, the modern concept of multi-dimensional rural development requires innovative tools that will fulfill its multiple purposes [70]. For this reason, assessing at the same time the ALUT, its driving factors, and agricultural land functions is an integrated approach that is less frequently used in previous scientific research in developing countries. In contrast to existing research that focuses on a single pattern, this theoretical approach, another contribution of this study, provides a more accurate portrayal of the spatial ALUT and its impact on living security and eco-environment maintenance in agriculture. In other words, from a theoretical perspective, this study broadens the research field on ALUT at a multi-dimensional level. It integrates other factors, such as the lack of agricultural investment and land tenure status. More specifically, this study has discovered several challenges concerning the nexus between the four factors of natural, climatic, socio-economic, and political driving factors in the Thiès region. Ultimately, the study's findings contribute to a better theoretical understanding of the factors influencing ALUT in Senegal through the case study in the Thiès region. In other words, the research contributes to the literature on land use transition by demonstrating the relations between agricultural land functions and influencing factors based on the survey statistics.

### 4.4. Policy Implications

Sustainable land use management in semi-arid agriculture requires quantitative and qualitative understanding and exploring their potential driving factors. Our study found that agricultural land use decreased rapidly between 2000 and 2020, and several driving factors have caused these decreases; accordingly, an integrated agricultural production system and land conservation are necessary. For this reason, significant implications have been highlighted in this study.

(1)  Rainfall variability directly reflects on the evolution of the sown land area. Therefore, different water systems must be mitigated and strengthened using suitable underground water while accounting for environmental issues. This integrated water use in agricultural production positively affected crop yield, increasing agricultural land functions—particularly per capita agricultural production.

(2)  In the Thiès region, about 81.37% of the respondents did not have formal documents. In this context, customary and modern land rights simultaneously govern land in Senegal. This situation induces many issues. On the one hand, land access became a source of conflict between government agencies regarding deliberation and, on the other hand, between users for their exploitation. Consequently, strong land reform policies in Senegal are urgent in light of the Thiès region's worrying urbanization of agricultural land [43]. So, reforming land tenure must cooperate with its complexity rather than attempt to substitute customary land practices.

(3)  Cooperatives can also contribute to the strength of agricultural investment. Based on this potential, one of this study's most important policy implications is that agricultural cooperatives should be generalized and integrated with small-scale farming households for maximum exploitation of agricultural land use. In Senegal, agriculture is still done traditionally [71–76], and technology use is almost non-existent. This study believes that improving agricultural infrastructures and enhancing agricultural input via cooperatives may positively impact agricultural production and farmers' income, avoiding agricultural land abandonment.

(4)  The topography in the Thiès region is complex and diverse. This complexity induces erosion and soil salinization, affecting agricultural land use in the investigation area. The consequence of these degradations led to the evolution of different land utilization patterns and practices; therefore, agricultural land morphology needs a quantitative assessment of agricultural land degradation. Consequently, conducting land surveys using new technologies has become an urgent need to improve and combat land degradation to improve agricultural land conservation in Senegal.

### 4.5. Research Limitations and Prospects

Senegal is a predominantly rural economy where rain-fed production systems are the critical drivers for economic growth. So, the depreciation of agricultural production and decreased sown land area may strongly impact the population's living security. Therefore, although the study highlighted significant insights for ALUT, it presented specific limitations that should be explored in future investigations. The agricultural land use behavior from 2000 to 2020 shows that the unused land area increased during this research. Additionally, ecological land decreased significantly during the same period. The mining industry in the Thiès region could become more and more developed. This situation probably negatively impacts the agriculture sector and necessitates comprehensible analysis. Added to these issues, forest resources, namely ecological land, were essential in the study area for maintaining ecosystem functions and landscape connectivity. The survey revealed that about 42.05% of respondents had abandoned agricultural land. So, evaluating agricultural land abandonment's spatial and temporal patterns will benefit food security and ecological balance. In addition, the study has highlighted new and significant issues regarding living security. This issue also needs a long-term analysis to better understand the grain transition and agricultural income per capita evolution. So, future research will specifically focus on the impact of these issues on the implementation of innovative directions in agricultural

land use conservation and agricultural moderation. In addition, indicators such as topography, road density, and slope degree should be included in future research to understandably explore the driving mechanisms of changes in agricultural land use in the Thiès region.

## 5. Conclusions

The core aim of this research was to assess the spatial characteristics of the agricultural land use transition (ALUT) in the Thiès region and identify its potential driving factors. Agriculture land use dominates the Thiès region's land use morphology, accounting for 48% of the total land area in 2020. From a quantitative point of view, agricultural land use decreased from 2000 to 2020 by $-588.66$ km$^2$. Grassland was the most critical land use type to have participated in this loss. In addition, the share of construction land was about 148.95 km$^2$ during the same period. The kernel density shows that the ALUT was negative in the south and positive in the north of the Thiès region. The local spatial autocorrelation analysis appears to be identical between communes ($p$-value 0.029), with the most significant results in the north (low-low, high-low) and south-west (high-high). According to survey data, the living security functions were deficient if we compared them to the industrial counties. The agricultural income per capita was about USD 167.18; agricultural land per capita was 0.37 hectares per person. As a result of the research's findings, the current factors influencing ALUT and its functions are rainfall variability, a lack of agricultural investment, land tenure, research and development, and transportation facility expansion. The combination of natural, socio-economic, and political driving factors induced the ALUT in the Thiès region from 2000 to 2020. In addition, rural agricultural land in the Thiès region simultaneously faced insufficient land tools and heavy urbanization. Overarchingly, the Senegalese agricultural land use system faces multiple and complex influencing factors. Consequently, it needs to transition toward a regulated and specialized system, recognized through merging customary and legal land tenure and the homogenization of agricultural subsidies.

**Author Contributions:** Conceptualization, B.F. and G.D.; methodology, B.F.; software, B.F.; validation, B.F., G.D., C.L., T.S., E.M. and R.X.; formal Analysis, B.F.; investigation, B.F.; resources, B.F. and G.D.; data curation, B.F.; writing—original draft preparation, B.F.; writing—review and editing, B.F., G.D., C.L., T.S., E.M. and R.X.; visualization, B.F. and G.D.; supervision, G.D.; project administration, B.F. and G.D. All authors have read and agreed to the published version of the manuscript.

**Funding:** This research was funded by the National Social Science Foundation of China, grand No. 21BJY209.

**Institutional Review Board Statement:** Not applicable.

**Informed Consent Statement:** Not applicable.

**Data Availability Statement:** Not applicable.

**Conflicts of Interest:** The authors declare no conflict of interest.

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
