# Peer review of "Assessing the Spatial Agricultural Land Use Transition in Thiès Region, Senegal, and Its Potential Driving Factors"

_land, doi:10.3390/land12040779_

Round 1

Reviewer 1 Report

Before the paper is published, the following questions should be addressed.

In Abstract, the contribution of this work was not presented well, for example, what do several insights mean?

The preface chapter is too weak to summarize previous research, and some newest related literature should be added, such as the literature recommended below, but not limited to this.

https://doi.org/10.1016/j.ecolind.2023.109926

https://doi.org/10.3390/land12020366

https://doi.org/10.1016/j.habitatint.2023.102744

Line 27, the first two “and” are suggested to be replaced with commas, the current sentence is too long and not friendly to readers.

Figure 1, a, what’s the value? Elevation or altitude?

Line 154, part of the titles are not in bold style.

Line 164, Landsat4-5 TM?

Line 167, determining should be “determine”.

Section 3.1.1, regarding the introduction of the source of remote sensing data, the attribute of the data is currently too little. For example, the number of periods of data, and the source time of each image, you can consider adding a table. As for how to select remote sensing data, there are relatively too many introductions.

Line, 203-204, It seems that supervised classification does not include a series of operations listed after "such as".

Line 225. It may be a wrong numbering system which is similar to the Line 34.

Moreover, it's also aligned wrong.

Line 278, the formula does not align with Formula (3-4). Typesetting should be uniform.

Line 350, This number is also wrong, it should not have the same level with the Line 348.

Line448-452, the typography of this paragraph is also wrong.

The results and analysis part needs to be further condensed.

Section 5.1-5.2, wrong typography.

5.2-5.4, it is recommended to separate this part of the content into a Section Discussion, before mentioning the Section Conclusion. This chapter is organized in the order of 5.3, 5.2, and 5.4.

Section 5.1, It is recommended to further mine the contribution of this work.

Line 681-682, References should be numbered 14.

Author Response

Dear reviewer,

Thereby, receive the responses to the suggestions for the paper assessing the spatial agricultural land use transition in Thies region, Senegal, and its potential driving factors. 

Best regards. 

Reviewer 2 Report

This paper is part of the theme related to the processes of rural-urban interaction and, more specifically, in an aspect related to de-agrarianization: the decrease in areas dedicated to agricultural production and the resulting problems (reduction in food production, impairment of ecosystem services, etc.). The specific objective of this research is to assess the spatial characteristics of the transition of agricultural land use in Senegal through the case study of the Thiès region and to identify its possible drivers. It is, therefore, an issue that is universal today and constitutes a global problem.

The article corresponds perfectly with the theme of Land magazine and is, in general, correct. However, it is convenient to highlight the practical non-existence of comparative references with similar processes in other territories, both in Africa and on other continents. Consequently, the article presents an excessively local character, without observing differences, similarities and contrasts with other rural-urban transition processes.

The text does not go into depth in the study of the factors and characteristics of the urbanization processes that are driving the loss of agricultural land in this specific area.

The authors insist much more on methodological aspects than on the essential interpretation and explanation of the processes.

Author Response

(The authors gave the same response as above.)

Reviewer 3 Report

1. The abstract is surprising since it provides the results of the regression, which are different from table 5 and the explanation in 5.

2 each time different variables are mentioned. 2. there is no discussion section in the paper to relate the findings to the theoretical statements. They just discuss the results in section 4.1

3. the English is not very smooth, for example line 42 factors treat agricultural lands??? Line 121 they speak of an agricultural country instead of a region

4. they suggest that I know what the construction land dynamics index is (line 57)

5. the result that agricultural land is replacing grassland is the result of a conflict between settlers and nomad tribesmen with herd, which is not unique for Senegal

6. what is a weak farm's labour force? line 298

7. section 4.1 and 4.2 have a different system of subsections

8. line 452 what means Globally the relationship between etc.??

9. had not instead of hadn't (line 461)

10. line 519 degradation while suitable use instead of or suitable use

11. using underground water (571) is using a non-renewable resource and has large consequences for the environment

12. line 580 starts all of sudden about cooperatives. Why are they introduced?

13 section 5.3and 5.4 need alignment left

My main problem is that most of the paper is about the sophisticated methods authors use, very little is about things that matter in this region, such as the high population growth, the lack of natural resources and the impact of the nearby capital Dakar, which doubled its population in the last 50 years!

Author Response

(The authors gave the same response as above.)

Round 2

Reviewer 1 Report

In response to the comments put forward, the authors carefully carried out the revision work point by point. The new version solved most issues now. I think the quality of the current edition has been greatly improved and this paper can be published.